# Increases in Ginsenoside Rg3, Compound K, and Antioxidant Activity of Cultivated Wild *Panax* Ginseng (CWPG) by Puffing

**DOI:** 10.3390/foods11192936

**Published:** 2022-09-20

**Authors:** Gwang-Su Choi, Jae-Sung Shin, Wooki Kim, Moo-Yeol Baik

**Affiliations:** Department of Food Science and Biotechnology, Institute of Life Science and Resources, Kyung Hee University, Yongin 17104, Korea

**Keywords:** cultivated wild *Panax* ginseng, antioxidant property, acidic polysaccharide, puffing, total phenolic contents (TPC), total flavonoid contents (TFC)

## Abstract

Cultivated wild *Panax* ginseng (CWPG) has been reported to have a higher content of ginsenoside than normal *Panax* ginseng. This study was carried out to increase the antioxidant activity and active ingredients by the puffing process. Therefore, effects of moisture content and pressure conditions on the antioxidant activity and active ingredients of CWPG were investigated. Extraction yield and crude saponin content were decreased at all moisture contents with increasing pressure. HPLC analysis showed that the contents of ginsenoside Rg3 and compound K were increased by puffing when the pressure increased. Antioxidant properties, total phenolic content (TPC) and total flavonoid content (TFC) were increased by puffing. The correlation between color change and antioxidant activity showed the greatest correlation with the decrease of L value. It is expected that the progress of this study will play an important role in the international market of high-value-added food using CWPG.

## 1. Introduction

*Panax* ginseng is known to contain saponin and non-saponin components, including phenolic compounds, flavonoids and polysaccharides, and has been used for medicinal applications in various diseases and symptoms over thousands of years. Specifically, the *Panax* ginseng polysaccharides were shown to have immune-modulating [1] and anticancer effects [2,3]. Furthermore, ginsenosides, the saponin components of ginseng, have been known to be beneficial in prevention/treatment of cardiovascular diseases [4,5], immune regulation [6], anticancer effects [5], and antidiabetic properties [7]. Due to the residual issues of artificial fertilizers or pesticides during cultivation, cultivated wild *Panax* ginseng (CWPG) was introduced by sowing ginseng seeds or wild ginseng seeds in mountain areas. With its higher price owing to the low yield in fertilizer- and/or pesticide-free environments, CWPG is generally known to have more potent pharmacological activities and stronger flavors as compared to the conventionally cultivated ginsengs. CWPG can be distinguished from conventional ginsengs by thinner and longer roots and a rounded band shape in the body [8].

For an elongation of storage time, ginseng is generally processed by repetition of drying and steaming, for which the heat contributes to (i) the breakdown of polymeric chemicals into more bioavailable small components and/or (ii) conversion of bioactive compounds into potent derivatives [9]. However, the conventional ginseng processing by steaming and drying is a time-consuming process over weeks and the soaking materials prior to the steaming are wasted, which all contribute to the cost rising. Therefore, it has been necessary to overcome these drawbacks by novel processing, yet taking advantages of the molecular alterations in a shorter time.

With respect to the processing methods, puffing is generally categorized into two types: oven puffing and gun puffing. The former utilizes the atmospheric pressure, while the later applies an additional pressure. Gun puffing is unique in that it is accompanied with both physical changes of the processed matrices and sterilization by heat and pressure applied. Briefly, a food-loaded chamber is heated, and steam and gases are generated inside the chamber, resulting in the rise of pressure. With the targeted pressure reached, the pressure drops to atmospheric status in an instant by opening of the chamber. Consequently, the internal moisture in food matrices evaporates due to the instantaneous decrease of pressure, making the food matrices porous [10,11]. During the puffing process, high pressure and temperature can contribute to the structural changes of the active compounds and a lower water content, which might aid in long-term storage. Due to their low-moisture and porous properties, puffed foods have an advantage of being easily pulverized, resulting in easier powdering for use in food materials.

Puffing causes changes not only in the physical properties but also in chemical characteristics of foods. The changes of ginsenoside profile [12,13,14], antioxidant activity [14], and aroma composition [12] in conventional ginsengs were confirmed in previous reports. Gelatinization of starch in cereal grains [15] and chemical alteration of isoflavones in soybeans [16] followed by puffing were also reported.

Puffed foods are generally brown-pigmented, which implies that heat- and pressure-added processing might affect the browning of carbohydrates. In this regard, it is well documented that the Maillard reaction, initiated by the condensation of an amine group in an amino acid and a carbonyl group in a monosaccharide, occurs in heat-processing of foods such as confectionery, bakery products, and grilled meat. As a consequence of complex chemical reactions, the end products, i.e., Maillard reaction products (MRPs), including brown-pigment melanoidin and volatile aromatic compounds, are generated [17]. Of interest, melanoidin has been reported to have a high antioxidant activity [18,19].

With those tentative puffing effects being known, the current study for the first time investigated the changes of active components in puffed CWPG prepared with various moisture contents and puffing pressures, seeking an optimal puffing condition for the maximal biofunctionality of antioxidant capacity.

## 2. Materials and Methods

### 2.1. Materials and Chemicals

Compositional and/or functional properties of natural products are affected by various factors including climate. Therefore, in the current study, cultivated wild *Panax* ginseng (CWPG) grown for 36 months in Pyeongchang, Korea was purchased from Woori Do (Pyeongchang, Korea). The CWPG was rinsed three times with running water to remove soil from the surface, and the surface was dried after the final rinsing with distilled water and stored in a −20 °C freezer.

Ethanol was purchased from Ethanol Supplies World Co. (Jeonju-si, Korea). Diethyl ether and n-butanol were purchased from Daejung Chemicals & Metals Co. (Siheung-si, Korea). Folin–Ciocalteu reagent, 2,2-diphenyl-1-picrylhydrazyl (DPPH), 2,2′-azobis-(2-amidinopropane) dihydrochloride (AAPH), 2,2′-azino-bis (3-ethylbenzothiazoline-6-sulphnic acid (ABTS), and phosphate-buffered saline (PBS) were purchased from Sigma-Aldrich (St. Louis, MO, USA). Other reagents used were of analytical grade.

### 2.2. Puffing Process

Three-year-old CWPG was cut to an approximate thickness of 1 cm and dried using a 40 °C hot air dryer (HB-502M, HanBeak Scientific Co., Bucheon-si, Korea) for adjustment of the final moisture content to 14, 10, or 8% (wet basis). For the optimal puffing condition by prevention of CWPG carbonization in high temperature, dry CWPG slices were physically mixed with rice at a ratio of 1:50 (*w*/*w*) in a bowl to prevent excessive carbonization, as previously reported [12,13,20], followed by heating in a gun puffing machine (Shinhak Food Machine, Seoul, Korea) until the pressure reached at 490 kPa. Subsequently the intermediate pressure was lowered to 294 kPa, followed by the pressure rising to 686, 784, 882, or 980 kPa by heating. At the designated internal pressure, the chamber of the puffer was instantly opened to induce the puffing of food materials. Non-puffed CWPG served as a control of the process.

### 2.3. Extraction

The puffed CWPG samples were ground, and 70% ethanol at a volume of 25 mL was added to 1 g of the solid content, followed by 30 min stirring at room temperature [12,13,20]. The extracts were further filtered in a Kimble-filtering flask (Sigma-Aldrich Co., St. Louis, MO, USA) with Whatman no. 2 filter paper. The extract was dried in a hot-air dryer (HB-502M, HanBeak Scientific Co., Bucheon-si, Korea) at 105 °C and the extraction yield was calculated using the following Equation (1).
(1)$$\mathrm{Extraction}\;\mathrm{yield}\;\left(\%\right)=\frac{\left(W_{2}-W_{1}\right)}{A}\times \frac{E}{E^{\prime }}\times 100$$
where

A stands for weight of sample (g)

E stands for total volume of extract (mL)

E′ stands for used volume of extract (mL)

W_1_ stands for initial weight of aluminum dish (g)

W_2_ stands for weight of aluminum dish and solid (g)

The filtrate was concentrated in a rotary vacuum evaporator (EYELA Rotary Vacuum Evaporator N-11, EYELA, Tokyo, Japan). Dried solids were dissolved in 5 mL of distilled water and aliquoted for the following assessments.

### 2.4. Crude Saponin Analysis

The crude saponin contents of the extracts were determined by the method of An et al. [13] by mixing 5 mL of concentrate, 20 mL of distilled water, and 25 mL of diethyl ether in a separatory funnel (Sigma-Aldrich Co., St. Louis, MO, USA). The mixture was shaken well and allowed to stand for 30 min until the separation of water and ether layers. Following the discarding of the separated ether layer, the remaining water layer was again mixed with 25 mL of water-saturated butanol. The mixture was further shaken and set to stand for separation for another 30 min. The separated water-saturated butanol layer was collected and the water layer was again washed with water-saturated butanol. Fifty milliliters of water was added to the separated water-saturated butanol, and the mixture was shaken well. Then, the mixture was allowed to stand until it was separated again, and the saturated butanol layer was evaporated using a rotary vacuum evaporator (BCHI rotavapor R-124, Flawil, Switzerland) for concentration under reduced pressure. After concentration under reduced pressure, it was dried at 105 °C in an oven for 2 h. The crude saponin content was calculated by weighing.
(2)$$\mathrm{Crude}\;\mathrm{saponin}\;\left(\mathrm{mg}/g\;\mathrm{ginseng}\right)=\frac{W_{1}-W_{2}}{W_{3}}\times \frac{A}{B}$$

W_1_: Weight of the dried sample and flask (mg)

W_2_: Weight of the flask (mg)

W_3_: Weight of total CWPG (g)

A: Weight of total concentrations (g)

B: Weight of used concentrations (g)

### 2.5. Ginsenoside Profiling

Ginsenoside content was determined by dissolving crude saponin in 5 mL of HPLC-grade methanol, followed by filtering with a Millipore filter (pore size 0.45 µm) and HPLC (Agilent 1260, Santa Clara, CA, USA) analysis. The instrument was equipped with Phenomenex Kinetex C18 (50 × 4.6 mm, ID 2.6 μm) and a UV detector at 203 nm. A binary gradient elution solvent consisting of distilled water (A) and acetonitrile (B) was used as mobile phase: 0–7 min, 81% A, 19% B; 7–11 min, 71% A, 29% B; 11–14 min, 71% A, 29% B; 14–25 min, 60% A, 40% B; 25–28 min, 44% A, 66% B; 28–30 min, 30% A, 70% B; 30–31.5 min, 10% A, 90% B; 31.5–34 min, 10% A, 90% B; 34–34.5 min, 81% A, 19% B; 34.5–40 min, 81% A, 19% B. The flow rate of the mobile phase was 1.0 mL/min, the sample injection amount was 5 μL, and the analysis temperature was 45 °C. A gradient elution procedure was used as follows.

### 2.6. Quantification of Antioxidant Activities

#### 2.6.1. DPPH Radical Scavenging Activity

The DPPH free radical scavenging activity of the extracts was measured by modifying the method of [21] using 2,2-diphenyl-1-picrylhydrazyl and 80% methanol to prepare a 0.1 mM DPPH solution. Briefly, DPPH solution and extracts were mixed and reacted at room temperature for 30 min. The absorbance at 517 nm subtracted by a blank absorbance was determined. Vitamin C served as a standard, for which the radical scavenging activity of the extract was expressed as mg vitamin C equivalent (VCE)/g dry ginseng.

#### 2.6.2. ABTS Radical Scavenging Activity

ABTS radical scavenging activity of the extracts was measured by modifying the method of [22]. Briefly, a mixture of 1.0 mM AAPH (2,2’-azobis- (2-amidinopropane) dihydrochloride), 2.5 mM ABTS (2,2’-azino-bis (3-ethylbenzothiazoline-6-sulphonic acid), and phosphate buffer saline was reacted for 30 min at 70 °C. After filtration through a 0.45 μm syringe filter, 980 uL of ABTS reagent was applied to 20 uL of CWPG extract and reacted at 37 °C for 10 min followed by the absorbance measurement and subtraction by a blank. Vitamin C served as a standard, for which the radical scavenging activity of the extract was expressed as mg vitamin C equivalent (VCE)/g dry ginseng.

### 2.7. Assessment of Total Phenolics and Total Flavonoids

Total phenolic contents (TPC) were measured by the modified Folin–Ciocalteu method [23]. Briefly, 200 μL of extract, 2.6 mL of distilled water, and 200 μL of Folin–Ciocalteu solution were reacted for 8 min. Following the reaction, 2 mL of Na_2_CO_3_ was added and absorbance was measured at 750 nm after another 82 min. Gallic acid served as a standard, for which total phenolic contents were expressed as mg Gallic acid equivalent (GAE)/g dry ginseng.

Total flavonoid content (TFC) was measured by modifying the method of [24]. Briefly, 0.5 mL of diluted extracts, 3.2 mL of distilled water, and 0.15 mL of 5% NaNO were mixed. After 30 min, 0.15 mL of 10% AlCl₃ was added. After 1 min, 1 mL of Na_2_CO_3_ was added and the absorbance at 510 nm was measured. Catechin served as a standard, for which total flavonoid contents were expressed as mg catechin equivalent (CE)/g dry ginseng.

### 2.8. Acidic Polysaccharide

Quantification of acidic polysaccharide was determined using a colorimetric method with some modifications (Do, Lee, Lee, Jang, Lee, & Sung, 1993). Three mL of H_2_O and 0.25 mL of 0.01% carbazole solution (in EtOH) were added to 0.5 mL of CWPG extract. The mixture was reacted for 5 min in a water bath (SB 1100, EYELA, Tokyo, Japan) at 85 °C, followed by cooling for 20 min at room temperature. Immediately, absorbance at 525 nm was measured using a spectrophotometer (UV-1200, Labentech, Incheon, Korea), and distilled water was used in the control group. D-galacturonic acid was used as a standard.

### 2.9. Color Measurement

In order to quantitatively analyze the color change of the puffed CWPG, the color difference of pulverized puffed CWPG was measured using a color difference meter (JC801, Color Techno System, Tokyo, Japan). All measurement was performed on a triplicate sample. Hunter L, a, and b values ranged from L = 0 (black) to 100 (white), a = −80 (greenness) to 100 (redness), and b = −80 (blueness) to 70 (yellowness). The calibration of the colorimeter was set with the standard plate with L = 98.26, a = 0.24, and b = −0.24.

To measure the level of Maillard reaction products, the extract was diluted 10-fold and measured at a wavelength of 420 nm using a spectrophotometer.

### 2.10. Statistical Analysis

All experiments were recapped three times and data were expressed as the mean ± standard deviation (SD). Experimental data were further analyzed by analysis of variance (ANOVA) and expressed as mean value ± standard deviation. Duncan’s multiple range test followed to assess significant differences among experimental mean values (*p* < 0.01 or *p* < 0.05). All statistical computations and analyses were conducted with SAS software (version 8.2, SAS Institute, Inc., Cary, NC, USA).

## 3. Results and Discussions

### 3.1. Extraction Yield and Crude Saponin Contents

The extraction yield and crude saponin content are shown in Table 1. The extraction yield tended to decrease following the puffing process. Specifically, compared to the yield of non-puffed control at 23.80 ± 0.40%, all puffed CWPG extracts exhibited significantly lower yields (*p* < 0.05) ranging from 18.42 ± 1.51 to 21.87 ± 1.01%. In addition, the more pressure applied in puffing, statistically the less extraction yield was observed in 14% and 8% moisture treatments; 27.72 ± 1.15 (686 kPa) to 20.01 ± 1.66% (980 kPa) for moisture content 14% and 21.87 ± 1.01 (686 kPa) to 18.59 ± 3.97% (980 kPa) for 8% moisture content. The decrement of extraction yield might be owed to the carbonization and/or physical alteration by heat treatment and puffing. There was no difference between the saponin contents of control vs. 14%, control vs. 10%, and control vs. 8%, at all levels of pressures. Within the 14% group, there was no difference between different pressure treatments. The same was true for the 10% group; however, within the 8% group, 686 kPa treatment clearly led to an increased saponin level when compared to 980 kPa.

The observation of decreased extraction yield, yet the comparable crude saponin content following puffing indicates that non-saponin components of CWPG are prone to thermal degradation. This observation is in contrast to the previous observation where increase of both extraction yield and crude saponin content was reported [12,13]. However, Yoon et al. (2010) reported no change in extraction yield of ginseng by puffing [25]. These observations indicate that the puffing condition and the puffed material significantly affect the extraction yield and crude saponin content. In the current study, the changes of these parameters following puffing of CWPG are reported for the first time.

### 3.2. Determination of Acid Polysaccharide Contents

A pile of studies indicated certain polysaccharide-containing carboxylic groups, termed acidic polysaccharides, exert the antioxidant properties in ginsengs [26]. Therefore, the contribution of puffing on acidic polysaccharide extraction in CWPG was also investigated, as shown in Table 1. The content of acidic polysaccharides in non-puffed CWPG extracts was 2.26 ± 0.75 mg galacturonic acid-equivalent/g dry ginseng. Following the puffing process, acidic polysaccharide contents varied from 12.76 to 18.97 mg galacturonic acid equivalent/g dry ginseng, implying that puffing does significantly affect the extraction of acid polysaccharides (*p* > 0.05). As the pressure treatment increased, the degree of decomposition of the acid polysaccharide tended to increase generally.

Following the method of Do et al. [27], the acidic polysaccharide content varied from 17.20 to 49.61 mg of galacturonic acid equivalent/g dry ginseng. This method is not suitable for measuring the degree of decomposition of the acidic polysaccharide by the puffing process. In contrast, Gui and Ryu [28] reported that the amount of acid polysaccharide increased with the increase of extraction yield by shear force, pressure, and heat in conventional ginsengs. The discrepancy might be caused by the decreased extraction yield (Table 1) following the puffing process. Alternatively, it was demonstrated that α-amylase in the ginseng contributed to the liberation of acidic polysaccharides during the steaming process, for which steam temperature was kept milder (<100 °C) than during puffing [29]. In contrast, puffing utilizes higher temperature and pressure for physicochemical alteration in relatively lower moisture contents, which might induce inactivation of α-amylase.

### 3.3. Effects of Puffing on Ginsenoside Contents

HPLC chromatograms of puffed and non-puffed CWPG are presented in Figure 1. Peaks in HPLC chromatograms of puffed CWPG in various puffing conditions are very comparable. Therefore, only representative HPLC chromatograms of non-puffed and puffed CWPG are presented in Figure 1 to visualize the changes in ginsenosides. Additionally, changes of ginsenosides in puffed and non-puffed CWPG are shown in Figure 2. The major ginsenosides, as grouped in the blue square, were decreased following puffing (Figure 1A), while minor ginsenosides shown in the red square were increased (Figure 1B). This suggests that puffing affects ginsenoside conversion. Panels A, B, and C in Figure 2 represent the CWPG extracts puffed with water content of 8, 10, and 14%, respectively. In general, alteration patterns in ginsenoside composition following puffing seem irrelevant to the water content. In the non-puffed control CWPG extract, major ginsenosides were in the following order; Rg1 > Re > Rf > Rb1 > Rc > Rb2 > Rd. Interestingly, those major ginsenoside contents were decreased by the puffing process in a pressure-dependent manner. Of most surprise was that the content of compound K (C-K) was dramatically increased by puffing while Rg3 was also slightly yet significantly increased in a puffing pressure-dependent manner. In accordance with the current observation, our laboratory has already reported that puffing increases the minor ginsenoside Rg3 in conventional *Panax* ginseng in a pressure-dependent manner [12]. These observations imply that the heat and pressure treatment of CWPG during the puffing process might bring about the conversion of ginsenosides. Health-beneficial effects of ginsenosides Rg3 and C-K have been extensively studied in previous reports, where anticancer effects [30] and immune modulation [31] were demonstrated. In this regard, Jung et al. reported that minor ginsenosides have better bioavailability and cellular uptake due to their smaller molecular sizes [32]. Similarly, thermal processes are widely applied to produce red or black ginseng, in which those minor ginsenosides are enriched [33]. Thus, the current study implies that puffing can replace time- and cost-consuming thermal processes, yet comparably alter ginsenoside in a short time and at a low cost. In contrast, the contents of Rg2 and Rh2 were relatively low in the control CWPG extracts and remained constant following puffing regardless of the increment of the pressure, indicating that the amount of conversion was an intermediate product in balance. Indeed, the reports by [34,35] support this hypothesis in that the aforementioned two ginsenosides were in the middle range of each reaction.

### 3.4. Antioxidant Properties, TPC, and TFC

Following the observation of changes in ginsenoside profiles, it was investigated whether the puffing-induced physicochemical alteration in CWPG affects the antioxidant capacities. Panels A and B in Figure 3 exhibit the increase of antioxidant properties in puffed CWPG as assessed by DPPH and ABTS radical scavenging capacity, respectively, in a puffing pressure-dependent manner. These results indicate that the extraction of antioxidant components was increased as the internal structure of CWPG was porosified by the chemical bond-breaking puffing process [36]. Total phenolic contents (TPC) and total flavonoid contents (TFC) as depicted in Figure 3C,D were assessed to confirm the increased extraction by puffing. Indeed, in TPC and TFC, the well-documented antioxidant components in the plants [37,38] were increased by puffing of CWPG. In accordance with this, it was demonstrated that TPC and TFC increased following thermal treatment of tomatoes [39]. The correlation of antioxidant effects with TPC and/or TFC was tested as shown in Table 2. DPPH and ABTS were highly correlated as determined by Pearson’s correlation coefficient (r) of 0.959 (*p* < 0.005). Of interest, TFC exhibited higher correlation to DPPH (r = 0.945, *p* < 0.005) and ABTS (r = 0.956, *p* < 0.005) radical scavenging activities. TPC also demonstrated significant correlation to DPPH (r = 0.989, *p* < 0.005) and ABTS (r = 0.959, *p* < 0.005). In addition, TPC and TFC were also highly correlated with r value of 0.976 (*p* < 0.005), indicating that elevated extraction of polyphenols and flavonoids contributes to the fortified antioxidant capacity [38].

### 3.5. Production of MRPs by Puffing Process

The colorimetric properties of puffed vs. non-puffed CWPG are shown in Table 3. As compared to control CWPG, puffing induced decrease in all parameters, i.e., L-, a-, and b-values, indicating the ‘darkening’ of CWPG. The more puffing pressure was applied, the stronger was the decrease in all values observed. It seems that the initial water content in CWPG does not affect the color change during the puffing process. The color difference (ΔE) compared to the non-puffed control was increased as the stronger puffing pressure was applied, indicating that the pressure was one of the controlling factors of ‘darkening’ of CWPG. Absorbance measured at 420 nm of MRPs in the non-puffed CWPG was 0.05 ± 0.02. The level of MRPs increased significantly with puffing and increased with increasing puffing pressure. The maximum value at each water content level was found at 980 kPa. When the puffing pressure is increased, the heat treatment time is prolonged, so that the generation of MRPs due to heat is increased.

It was well introduced that during heat and pressure treatments, brown pigments, named Maillard reaction products (MRPs) including melanoidin, are formed in various foods, and consequently contribute to its antioxidant ability [18,19]. Therefore, the effects of color changes as quantified by L-, a-, b-values, and ΔE on the antioxidant capacity were determined using the aforementioned Pearson’s correlation analysis (Table 2). Interestingly, lightness of puffed CWPG, as quantified by L-values, exhibited a high inverse correlation with DPPH (r = −0.963, *p* < 0.005), ABTS (r = −0968, *p* < 0.005) radical scavenging activities, TFC (r = −0.603, *p* < 0.05), and TPC (r = −0.961, *p* < 0.005), indicating that L-values might be used as an indicator in puffing of CWPG. The ΔE value was highly correlated to DPPH (r = 0.953, *p* < 005), ABTS (r = 0.963, *p* < 0.005), TFC (r = 0.627, *p* < 0.05), and TPC (r = 0.946, *p* < 0.005). Maillard reaction products value was also highly correlated to DPPH (r = 0.967, *p* < 005), ABTS (r = 0.957, *p* < 0.005), TFC (r = 0.583, *p* < 0.05), and TPC (r = 0.961, *p* < 0.005). These results indicate that puffing of CPWG increases antioxidant capacities of the extracts.

## 4. Conclusions

The current study sought, for the first time, the effects of puffing on physicochemical changes in CWPG. Extraction yield and crude saponin content of CWPG were decreased when the initial water content was controlled. With respect to the profiles of ginsenoside, Rg1, Re, Rf, Rb1, Rc, Rb2, and Rd were decreased in the front part of the change mechanism, and the intermediate products Rg2 and Rh2 were kept constant. However, the final products, Rg3 and compound K, showed dramatic incremental changes following the puffing process. Subsequently, antioxidant activities, as determined by DPPH and ABTS radical scavenging, were increased with the increase of puffing pressure. In accordance, the content of total phenolic compounds and total flavonoids increased, which may result from the destruction of polyphenolic herbal matrices by puffing. As the puffing pressure increases, which was controlled by heating time, the L-, a-, and b-values of the CWPG were inversely correlated, consequently increasing the color difference ΔE to the non-puffed controls. Correlation analysis revealed that antioxidant activity was strongly correlated with L-value, i.e., darkness of the puffed CWPG. The acidic polysaccharide did not change significantly with puffing, but it seemed to be more sensitive to the decrease of extraction yield than to elution by puffing. In addition, it is assumed that the extraction conditions did not sufficiently extract the acidic polysaccharide. These observations support the tentative application of puffing to CWPG for the development of food materials, replacing the time- and money-consuming conventional steaming and drying methods.

## Figures and Tables

**Figure 1 foods-11-02936-f001:**
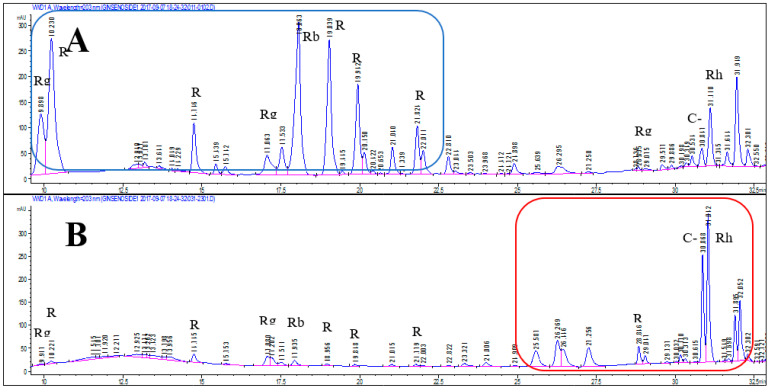
HPLC chromatograms of (**A**) non-puffed CWPG and (**B**) puffed CWPG. Representative histogram of puffed CWPG at 10% moisture content and 784 kPa puffing pressure is presented to visualize changes of peaks.

**Figure 2 foods-11-02936-f002:**
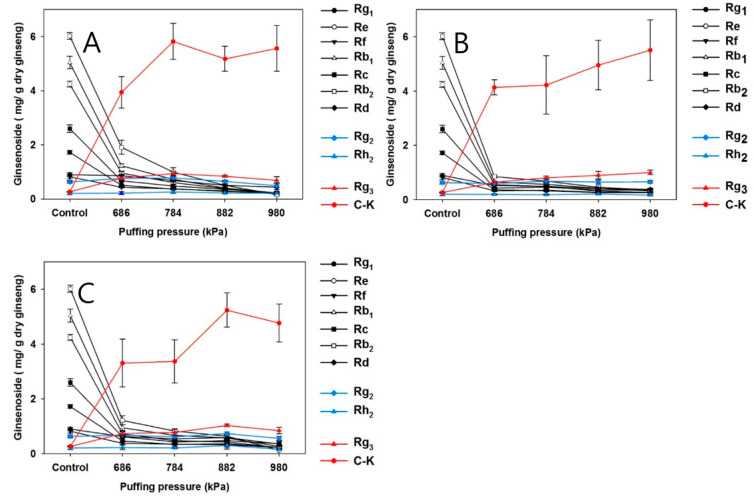
Changes in ginsenoside contents of puffed CWPG with different puffing pressure and moisture contents: (**A**) 8% moisture content, (**B**) 10% moisture content, (**C**) 14% moisture content.

**Figure 3 foods-11-02936-f003:**
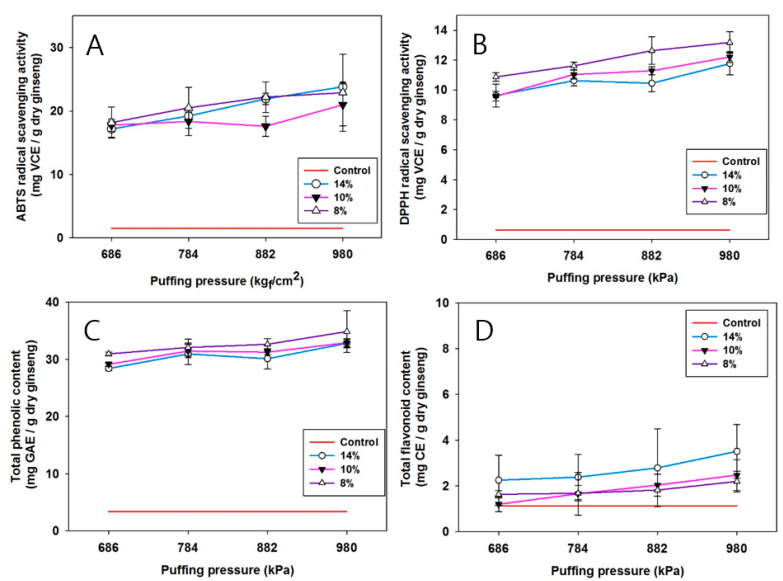
Effects of puffing pressure and water content on ABTS radical scavenging activity (**A**), DPPH radical scavenging activity (**B**), total phenolic content (**C**), and total flavonoid content (**D**) of puffed CWPG.

**Table 1 foods-11-02936-t001:** Extraction yields, crude saponin contents, and acidic polysaccharide contents of puffed CWPG with different puffing pressure and moisture contents.

Moisture Contents (%)	Puffing Pressure (kPa)	Extraction Yield (%)	Crude Saponin Content (mg/g Dried Ginseng)	Acidic Polysaccharide Contents (mg Galacturonic Acid Equivalent/g Dry Ginseng)
Control		23.80 ± 0.40 ^a^*	73.22 ± 7.42 ^ABCD^	2.26 ± 0.75 ^h^
14%	686	21.72 ± 1.15 ^bc^	70.28 ± 3.53 ^BCD^	12.87 ± 1.44 ^g^
	784	20.92 ± 1.79 ^bcd^	57.57 ± 7.63 ^D^	16.77 ± 2.50 ^bc^
	882	18.67 ± 1.58 ^ef^	69.73 ± 4.92 ^BCD^	15.35 ± 1.96 ^de^
	980	20.01 ± 1.66 ^cdef^	70.60 ± 6.51 ^BCD^	15.81 ± 1.95 ^cde^
10%	686	20.05 ± 2.59 ^cdef^	73.05 ± 4.98 ^ABCD^	14.94 ± 0.57 ^ef^
	784	20.32 ± 2.48 ^bcde^	60.28 ± 3.83 ^CD^	17.58 ± 0.95 ^b^
	882	18.42 ± 1.51 ^f^	68.42 ± 3.78 ^BCD^	13.26 ± 1.51 ^g^
	980	19.54 ± 1.33 ^def^	61.92 ± 3.06 ^CD^	16.22 ± 0.75 ^cd^
8%	686	21.87 ± 1.01 ^b^	89.45 ± 7.66 ^A^	12.76 ± 2.10 ^g^
	784	20.31 ± 0.72 ^bcde^	76.70 ± 7.50 ^ABC^	18.97 ± 2.01 ^a^
	882	19.17 ± 0.85 ^def^	83.90 ± 8.76 ^AB^	13.94 ± 1.07 ^fg^
	980	18.59 ± 3.97 ^ef^	69.63 ± 5.34 ^BCD^	17.43 ± 0.64 ^b^

* Values with the same letter in the same column are not significantly different (*p* < 0.05).

**Table 2 foods-11-02936-t002:** Pearson correlations between physicochemical properties of puffed CWPG.

	DPPH	ABTS	TFC	TPC	L	a	b	∆E	MRPs
DPPH		0.960 ***	0.465	0.989 ***	−0.963 ***	−0.677 *	−0.807 ***	0.953 ***	0.967 ***
ABTS			0.613 *	0.959 ***	−0.968 ***	−0.728 ***	−0.852 ***	0.963 ***	0.957 ***
TFC				0.467	−0.603 *	−0.728 ***	−0.723 **	0.627 *	0.583 *
TPC					−0.961 ***	−0.618 *	−0.768 ***	0.946 ***	0.961 ***
L						0.795 ***	−0.906 ***	−0.998 ***	−0.987 ***
a							0.974 ***	−0.828 ***	−0.776 **
b								−0.929 ***	−0.892 ***
∆E									0.985 ***
MRPs									

* *p* < 0.05, ** *p* < 0.01, *** *p* < 0.005.

**Table 3 foods-11-02936-t003:** Color and Maillard reaction products (MRPS) of puffed CWPG with different puffing pressure and moisture contents.

Moisture Contents (%)	Puffing Pressure (kPa)	L	a	b	∆E	MRPS ** (Absorbance at 420 nm)
Control		68.32 ± 1.56 ^a^*	9.33 ± 0.95 ^a^	25.91 ± 1.82 ^a^*		0.05 ± 0.02 ^H^
14%	686	36.53 ± 1.61 ^b^	8.32 ± 0.92 ^b^	19.23 ± 1.69 ^b^	32.49529	0.61 ± 0.04 ^G^
	784	30.01 ± 1.16 ^d^	6.94 ± 0.85 ^c^	14.64 ± 1.96 ^d^	40.00235	0.70 ± 0.02 ^F^
	882	25.48 ± 3.91 ^e^	5.35 ± 1.43 ^de^	9.94 ± 4.16 ^ef^	45.88677	0.77 ± 0.08 ^DE^
	980	22.32 ± 2.05 ^f^	4.46 ± 0.58 ^f^	7.27 ± 1.52 ^g^	49.87165	0.94 ± 0.01 ^A^
10%	686	32.64 ± 2.79 ^c^	7.88 ± 1.10 ^b^	16.68 ± 2.92 ^cd^	36.88232	0.69 ± 0.02 ^F^
	784	32.26 ± 0.76 ^c^	7.71 ± 0.40 ^bc^	16.51 ± 0.39 ^cd^	37.29976	0.74 ± 0.02 ^EF^
	882	25.18 ± 2.68 ^e^	5.46 ± 0.68 ^de^	10.63 ± 2.24 ^e^	45.93355	0.85 ± 0.02 ^BC^
	980	22.98 ± 1.32 ^f^	4.70 ± 0.45 ^ef^	8.39 ± 1.01 ^fg^	48.82999	0.90 ± 0.05 ^AB^
8%	686	34.71 ± 3.19 ^b^	8.27 ± 0.65 ^b^	17.85 ± 1.91 ^bc^	34.57783	0.72 ± 0.03 ^F^
	784	29.91 ± 1.10 ^d^	7.08 ± 0.30 ^c^	14.77 ± 0.88 ^d^	40.05783	0.80 ± 0.03 ^D^
	882	26.11 ± 1.15 ^e^	5.84 ± 0.32 ^d^	11.68 ± 0.79 ^e^	44.6799	0.82 ± 0.01 ^CD^
	980	22.42 ± 1.89 ^f^	4.43 ± 0.81 ^f^	7.99 ± 1.95 ^fg^	49.51781	0.87 ± 0.10 ^B^

* Values with the same letter in the same column are not significantly different (*p* < 0.05). ** MRPS: Maillard reaction products.

## Data Availability

Data is contained within the article.

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
