# Peer review of "Increases in Ginsenoside Rg3, Compound K, and Antioxidant Activity of Cultivated Wild Panax Ginseng (CWPG) by Puffing"

_foods, 2022, doi:10.3390/foods11192936_

Round 1
Reviewer 1 Report
The manuscript deals with evaluation of the effects of puffing on active components of cultivated wild panax ginseng.
In my opinion the experimental design is appropriate but results presentation and especially discussion can be improved. In addition, some mistake are present in the text.
Major revision
- Table1: It is not clear why the polysaccharide acid content is present in table 1, in the paragraph 3.1 of the results, but discussed in in another paragraph. I suggest to gather in the first paragraph or alternatively to separate the data into 2 different tables.
- Table 1: The authors should explain why the standard deviation is so high for the saponin content.
-Lines 222-227: Is the decreased of extraction yield due to wild Panax gingeng and its different composition compared to not wild ginseng? The authors should better explain and give a justification of the data obtained.
- Fig.1: Authors should explain why they used sample moisture contents 10%, puffing pressure 784 kPa
- Fig.2 line 262: The legend refers to figure 1 not 2, it must be modified.
- Paragraph 3.2: The authors declare that major ginsenoside contents were decreased by puffing process while the content of compound K (C-K) and Rg3 increased. Only C and BR have ealth-beneficial effects or do the others ginsenoside as well? The discussion needs to be improved.
- Lines 269-271: The sentence is not clear. Please clarify.
-Table 3: Why table 2 before table 3?
-Lines 298-299: The sentence is not clear. Please clarify.
-Lines 231: Are the Authors sure of the declaration? Is the phenomenon pressure dependent?
-Lines 332-337: It is not possible to compare the results obtained by the authors with those of other trials in which the extraction yield increased.
-Lines 351-358: Justify the sentences.
-Lines 358-360: Why didn't the authors use better extraction conditions?
In addition, some results are dependent pressure and others are not. Authors should discuss and justify this statement.
Minor revision
Line 235: The point is a mistake
Line 311 and 334: The double point is a mistake
Author Response
Reviewer 1
The manuscript deals with evaluation of the effects of puffing on active components of cultivated wild panax ginseng.
In my opinion the experimental design is appropriate but results presentation and especially discussion can be improved. In addition, some mistake are present in the text.
Major revision
- Table1: It is not clear why the polysaccharide acid content is present in table 1, in the paragraph 3.1 of the results, but discussed in in another paragraph. I suggest to gather in the first paragraph or alternatively to separate the data into 2 different tables.
|
Thank you for your comment. The authors agree that the polysaccharide acid content should be discussed following extraction yield and crude saponin contents. Therefore the paragraph was moved to 3.2. |
- Table 1: The authors should explain why the standard deviation is so high for the saponin content.
|
Thank you for your comment. The current study utilized commercial cultivated wild Panax ginseng, and the compositional variation (standard deviation) of certain components in 1-10% of mean is frequently observed as reported in Shin et al., Journal of Microbiology and Biotechnology 29(2): 222-229, 2019 |
-Lines 222-227: Is the decreased of extraction yield due to wild Panax gingeng and its different composition compared to not wild ginseng? The authors should better explain and give a justification of the data obtained.
|
Thank you for your comment. The current study did not compare the extraction yield of CWPG to conventional Panax ginseng. Instead, the extraction yield of CWPG after puffing process was reported to decrease, and the tentative mechanism of carbonization was described in line 234-236 of revised manuscript. |
- Fig.1: Authors should explain why they used sample moisture contents 10%, puffing pressure 784 kPa
|
Thank you for your comment. Peaks in histograms of puffed CWPG at various puffing conditions are very comparable. Therefore, only representative histogram was presented in the figure, and the figure legend was revised appropriately. |
- Fig.2 line 262: The legend refers to figure 1 not 2, it must be modified.
|
Thank you for your comment. We changed it accordingly. Please see line 274-278 of revised manuscript. |
- Paragraph 3.2: The authors declare that major ginsenoside contents were decreased by puffing process while the content of compound K (C-K) and Rg3 increased. Only C and BR have health-beneficial effects or do the others ginsenoside as well? The discussion needs to be improved.
|
Thank you for your comment. The health beneficial effects of minor ginsenosides were described in line 293-295 of revised manuscript. |
- Lines 269-271: The sentence is not clear. Please clarify.
|
Thank you for your comment. The sentences are revised appropriately in line 293-298 of revised manuscript. |
-Table 3: Why table 2 before table 3?
|
Thank you for a keen comment. The authors agree that table 2 and 3 should be reordered by their appearance in the text. |
-Lines 298-299: The sentence is not clear. Please clarify.
|
Thank you for your comment. The sentences were revised appropriately in line 314-322 of revised manuscript. |
-Lines 331: Are the Authors sure of the declaration? Is the phenomenon pressure dependent?
|
Thank you for your comment. Although extraction yield was not positively dependent on pressure level, the increase in MRPs concentration was proportionally dependent on pressure. |
-Lines 332-337: It is not possible to compare the results obtained by the authors with those of other trials in which the extraction yield increased.
|
Thank you for your comment. As mentioned above, extraction yield was not consistent with pressure level but MRPs increased with increasing puffing pressure in this study. Increase in extraction yield with puffing could not be always true. No significant increase in extraction yield of puffed Curcuma longa L. (Tumeric) has been reported by Choi et al., Antioxidants, 8, 506, 2019. |
-Lines 351-358: Justify the sentences.
|
Thank you for your comment. The sentences were revised appropriately in line 358-362 of revised manuscript. |
-Lines 358-360: Why didn't the authors use better extraction conditions?
|
Thank you for your comment. The current study focuses on the effect of puffing for increment of health-beneficial components. Therefore, the fine-tuning of extraction conditions is a separate scope for successive studies. |
In addition, some results are dependent pressure and others are not. Authors should discuss and justify this statement.
|
Thank you for your comment. As mention earlier, all properties are not increased with puffing. For example, major ginsenosides and some polysaccharides are disintegrated by puffing. By those disintegration, minor ginsenosides, acidic polysacchrides, and MRPs are increased. We discuswsed these things in the revised manuscript. |
Minor revision
Line 235: The point is a mistake
|
Thank you for your comment. We changed it accordingly. |
Line 311 and 334: The double point is a mistake
|
Thank you for your comment. We changed it accordingly. |

Reviewer 2 Report
The authors investigated the effects of puffing on the ginsenoside profiles, total phenolic/flavonoid contents, antioxidant activity, acidic polysaccharide contents, and colors of cultivated wild Panax ginseng (CWPG). Based on the data collected, the authors proposed the feasibility of puffing in the processing of CWPG. Strength of the study is that the authors have adopted different methodologies (chromatography, spectrophotometric assays, colour measurement, etc.) in their study.
Below are my feedbacks for the authors’ consideration during revision:
1. The manuscript seems not carefully checked prior to submission. For example,
· An identical caption “Figure 1. HPLC chromatograms of (A) non-puffed CWPG and (B) puffed CWPG (moisture contents 10%, puffing pressure 784 kPa).” has been used for two set of figures. Please see lines 258-259 and lines 262-263.
· The text mentions Fig. 2 (line 234). But the manuscript has no Figure 2 due to the caption error.
· Table 3 appears before Table 2. Please see lines 283 and 304.
· Lines 205 – 219 : all of the necessary ± (plus/minus) symbols are missing.
2. The first part of the paper investigated the ginsenoside profiles, then the second part switched to investigating antioxidant activity and total phenolic/flavonoid contents. It is unclear how ginsenoside profiles were connected to antioxidant activity.
3. Since this is the first time ginsenoside profiles in puffed CWPG were investigated, it would be good if any novelty in the findings can be explicitly highlighted.
4. Line 55 – is there a missing word after “atmospheric”?
5. In M&M, lines 81-84: to ensure reproducibility of results if the same study were to be repeated in future, it would be good if the authors provide more specific details on their samples. For example, size, time/years of growth, the “specific region” where cultivation happened, etc.
6. Line 94: The “three-year-old” description refers to exactly 36 months? Or is there a range?
7. Line 97: It would be clearer if the authors can briefly explain the step of mixing the sample with rice. In addition, how was the ratio 1:50 established?
8. Lines 95 and 100: How did the authors establish that 14, 10, 8% moisture and 686, 784, 882, or 980 kPa were appropriate treatments to be investigated? For example, why not 15%, instead of 14%? Why not 670 kPa instead of 686 kPa? Brief clarification in the text will make it clearer to readers.
9. Line 104: Is there a justification why extraction should use 70% ethanol rather than other solvent and percentage?
10. Generally, in M&M, especially sections 2.2 – 2.5, the authors did not cite any references for the procedures. Did they create/invent all those procedures?
11. The authors measured acid polysaccharides (M & M, line 178). Could the authors provide a briefing explanation in the text why neural polysaccharide was not of interest?
12. RESULTS and DISCUSSION:
· Lines 213 – 215: “With respect to crude saponin contents, the decreasing tendency … though the statistical power lacks (P>0.05).” – If it is not statistically significant, it would be inaccurate/meaningless to say they show a decreasing trend.
· Lines 215-219: This part seems to just repeat the same info already shown in the Table and appears unclear. Looking at statistical annotations in Table 1, I would suggest that there was no difference between the saponin contents of control vs 14%, control vs 10%, and control vs 8%, at all levels of pressures. Within the 14% group, there was no difference between different pressure treatments. The same goes for the 10% group. However, within the 8% group, 686 kPa treatment clearly led to increased saponin level when compared to 980 kPa.
· Lines 269-272: “… elution of antioxidant substances was increased as the internal structure of CWPG was porosified by chemical bond breaking puffing process. This hypothesis was tested, in part, by quantifying the eluted total phenolic contents (TPC) and total flavonoid contents (TFC)…” – I find this part unclear. The suggestion about “internal structure of CWPG was porosified by chemical bond breaking puffing” seems unjustified/unsupported by the authors own data/findings from other studies. It is also unclear to me how the hypothesis about increased porosity of CWPG can be tested by measuring TPC and TFC. Could the authors revise/elaborate this part?
· In M & M (2.11. Statistical analysis), the authors only mentioned P < 0.05 as a measure of statistical significance. But here in Table 3 (which should be Table 2???), the authors also considered P<0.01, and P<0.005. The authors may wish to consider revising section 2.11.
· Table 2 (which should be Table 3???): Info about statistical significance and SD is missing from the column of ∆E. Please recheck.
13. CONCLUSION: I think it would wrap up the paper nicely if the authors could also indicate whether the antioxidant activity was correlated to the change in ginsenoside profiles. If not, as mentioned above, the paper seems to be disconnected where the first part is about ginsenoside profiling, second part is mainly about antioxidant activity/phenolic/flavonoid content.
Author Response
Reviewer 2
The authors investigated the effects of puffing on the ginsenoside profiles, total phenolic/flavonoid contents, antioxidant activity, acidic polysaccharide contents, and colors of cultivated wild Panax ginseng (CWPG). Based on the data collected, the authors proposed the feasibility of puffing in the processing of CWPG. Strength of the study is that the authors have adopted different methodologies (chromatography, spectrophotometric assays, colour measurement, etc.) in their study.
Below are my feedbacks for the authors’ consideration during revision:
- The manuscript seems not carefully checked prior to submission. For example,
- An identical caption “Figure 1. HPLC chromatograms of (A) non-puffed CWPG and (B) puffed CWPG (moisture contents 10%, puffing pressure 784 kPa).” has been used for two set of figures. Please see lines 258-259 and lines 262-263.
- The text mentions Fig. 2 (line 234). But the manuscript has no Figure 2 due to the caption error.
|
Thank you for your comment. The figure legends were revised appropriately. |
Table 3 appears before Table 2. Please see lines 283 and 304.
|
Thank you for your comment. Following the reviewer’s comments, Table 2 and 3 were reordered as their appearance in the text. |
Lines 205 – 219 : all of the necessary ± (plus/minus) symbols are missing.
|
Thank you for your comment. We changed it accordingly. |
- The first part of the paper investigated the ginsenoside profiles, then the second part switched to investigating antioxidant activity and total phenolic/flavonoid contents. It is unclear how ginsenoside profiles were connected to antioxidant activity.
|
Thank you for your comment. As mentioned in Introduction, we investigated the changes of active components in puffed CWPG, which includes saponin, ginsenosides, TPC, TFC and MRPs. These may be interacted each other but not in some case. Therefore, we did not investigated the relationships between ginsenosides and antioxidant activity, because there are too many ginsenosides in puffed CWPG to check the interaction between them and antioxidant activity. |
- Since this is the first time ginsenoside profiles in puffed CWPG were investigated, it would be good if any novelty in the findings can be explicitly highlighted.
|
Thank you for your comment. The change of ginsenoside in puffed CWPG was examined in the current study as described by the reviewer. The results show that minor ginsenosides, which aid in the health-beneficial effects, are increased by puffing. This novel finding was highlighted in line 29-30 and 283-287 of revised manuscript. |
- Line 55 – is there a missing word after “atmospheric”?
|
Thank you for your comment. The sentence was revised appropriately in line 67 of revised manuscript. |
- In M&M, lines 81-84: to ensure reproducibility of results if the same study were to be repeated in future, it would be good if the authors provide more specific details on their samples. For example, size, time/years of growth, the “specific region” where cultivation happened, etc.
|
Thank you for your comment. The cultivation time of 36 months and specific area of Pyeongchang, Korea, were specified in line 94 of revised manuscript. |
- Line 94: The “three-year-old” description refers to exactly 36 months? Or is there a range?
|
Thank you for your comment. The growth time of 36 months were specified in line 94 of revised manuscript. |
- Line 97: It would be clearer if the authors can briefly explain the step of mixing the sample with rice. In addition, how was the ratio 1:50 established?
|
Thank you for your comment. The mixing of sliced CWPG and rice in a bowl was described in line 109 of revised manuscript. The rationale for 1:50 mixing was justified by a reference 12, 13, 20. |
- Lines 95 and 100: How did the authors establish that 14, 10, 8% moisture and 686, 784, 882, or 980 kPa were appropriate treatments to be investigated? For example, why not 15%, instead of 14%? Why not 670 kPa instead of 686 kPa? Brief clarification in the text will make it clearer to readers.
|
Thank you for your comment. The puffing conditions for conventional ginseng was set in a previous study as cited in references 12, 13, 20 and 36. Actually, the pressure gauge of puffing machine is set to 7, 8, 9, and 10 kgf/cm2 and we have to change it to kPa according to guide for authors in “Foods”. |
- Line 104: Is there a justification why extraction should use 70% ethanol rather than other solvent and percentage?
|
Thank you for your comment. Generally, 70% ethanol extraction is recognized as best extraction condition for both hydrophilic and hydrophobic materials in foods. Therefore, 70% ethanol extraction is widely used in ginseng industry and we followed this extraction method accordingly. |
- Generally, in M&M, especially sections 2.2 – 2.5, the authors did not cite any references for the procedures. Did they create/invent all those procedures?
|
Thank you for your comment. We add the appropriate references as you suggested. Please see lines 110, 118, 137 and 158 in the revised manuscript. |
- The authors measured acid polysaccharides (M & M, line 178). Could the authors provide a briefing explanation in the text why neural polysaccharide was not of interest?
|
Thank you for your comment. Previous studies indicate that acid polysaccharides, not neutral polysaccharides, are beneficial in health promotion. It is highlighted in the results of revised manuscript in line 254-257. |
- RESULTS and DISCUSSION:
Lines 213 – 215: “With respect to crude saponin contents, the decreasing tendency … though the statistical power lacks (P>0.05).” – If it is not statistically significant, it would be inaccurate/meaningless to say they show a decreasing trend.
|
Thank you for your comment. The authors agree that no statistical significance should be deemphasized in the text. |
Lines 215-219: This part seems to just repeat the same info already shown in the Table and appears unclear. Looking at statistical annotations in Table 1, I would suggest that there was no difference between the saponin contents of control vs 14%, control vs 10%, and control vs 8%, at all levels of pressures. Within the 14% group, there was no difference between different pressure treatments. The same goes for the 10% group. However, within the 8% group, 686 kPa treatment clearly led to increased saponin level when compared to 980 kPa.
|
Thank you for your comment. The authors appreciate for the reviewer’s thoughtful comments. The manuscript was revised appropriately. |
Lines 269-272: “… elution of antioxidant substances was increased as the internal structure of CWPG was porosified by chemical bond breaking puffing process. This hypothesis was tested, in part, by quantifying the eluted total phenolic contents (TPC) and total flavonoid contents (TFC)…” – I find this part unclear. The suggestion about “internal structure of CWPG was porosified by chemical bond breaking puffing” seems unjustified/unsupported by the authors own data/findings from other studies.
|
Thank you for your comment. The increment of porosity in conventional ginseng by puffing process was observed in a previous study, and cited appropriately in line 311-314 of revised manuscript. |
It is also unclear to me how the hypothesis about increased porosity of CWPG can be tested by measuring TPC and TFC. Could the authors revise/elaborate this part?
|
Thank you for your comment. The sentences are revised appropriately in line 316-324 of revised manuscript. |
In M & M (2.11. Statistical analysis), the authors only mentioned P < 0.05 as a measure of statistical significance. But here in Table 3 (which should be Table 2???), the authors also considered P<0.01, and P<0.005. The authors may wish to consider revising section 2.11.
|
Thank you for your comment. The statistical significance was defined at either p < 0.01 or p < 0.05, and the material and methods were revised appropriately in line 221-226 of revised manuscript. |
Table 2 (which should be Table 3???): Info about statistical significance and SD is missing from the column of ∆E. Please recheck.
|
Thank you for your comment. Table 2 and 3 were reordered in their appearance in the text. In addition, ∆E is calculated from the means of non-puffed control vs puffed CWPG as described in line 331-333 of revised manuscript. Therefore, ∆E values don’t have standard deviation. |
- CONCLUSION: I think it would wrap up the paper nicely if the authors could also indicate whether the antioxidant activity was correlated to the change in ginsenoside profiles. If not, as mentioned above, the paper seems to be disconnected where the first part is about ginsenoside profiling, second part is mainly about antioxidant activity/phenolic/flavonoid content.
|
Thank you for your comment. As mentioned above, we investigated the changes of active components in puffed CWPG, which includes saponin, ginsenosides, TPC, TFC and MRPs. Therefore, we did not investigated the relationships between ginsenosides and antioxidant activity, because there are too many ginsenosides in puffed CWPG to check the interaction between them and antioxidant activity.The current study finds that the changes in ginsenosides and antioxidant activity of puffed CWPG, separately. This finding was highlighted in line 360-364 of revised manuscript. |
